# `sympl` (v. 0.4.0) and `climt` (v. 0.15.3) – Towards a flexible framework for building model hierarchies in Python

Joy Merwin Monteiro[1], Jeremy McGibbon[2], and Rodrigo Caballero[1]

[1]Department of Meteorology, Stockholm University, SE-106 91 Stockholm, Sweden
[2] 408 Atmospheric Sciences–Geophysics (ATG) Building, University of Washington at Seattle, Box 351640, Seattle, Washington 98195-1640

**Correspondence:** Joy Merwin Monteiro (joy.merwin@gmail.com)

**Abstract.** `sympl` (System for Modelling Planets) and `climt` (Climate Modelling and Diagnostics Toolkit) are an attempt to rethink climate modelling frameworks from the ground up. The aim is to use expressive data structures available in the scientific Python ecosystem along with best practices in software design to allow scientists to easily and reliably combine model components to represent the climate system at a desired level of complexity, and to enable users to fully understand what the model is doing.

`sympl` is a framework which formulates the model in terms of a "state" that gets evolved forward in time or modified within a specific time by well-defined components. `sympl`'s design facilitates building models that are self-documenting, highly interoperable, and that provide fine-grained control over model components and behaviour. `sympl` components contain all relevant information about the input they expect and output that they provide. Components are designed to be easily interchanged, even when they rely on different units or array configurations. `sympl` provides basic functions and objects which could be used in any type of Earth system model.

`climt` is an Earth system modelling toolkit that contains scientific components built using `sympl` base objects. These include both pure Python components and wrapped Fortran libraries. `climt` provides functionality requiring model-specific assumptions, such as state initialisation and grid configuration. `climt`'s programming interface designed to be easy to use and thus appealing to a wide audience.

Model building, configuration and execution are performed through a Python script (or Jupyter Notebook), enabling researchers to build an end-to-end Python-based pipeline along with popular Python data analysis and visualisation tools.

## 1 Introduction

The climate is a complex system composed of interacting subsystems (atmosphere, ocean, cryosphere, biosphere, chemosphere), each encompassing a broad range of physical and chemical processes with space and time scales spanning many orders of magnitude. Modelling and understanding the climate system in its entirety is a grand scientific and technological challenge. An increasingly recognised strategy for tackling this challenge is to build a hierarchy of models of varying complexity. Simpler models are more amenable to in-depth analysis and understanding; the insight gained from these simpler models can then be used to understand more complicated models, and so on (Held, 2005). Specifying which particular models should

occupy each rung in such a hierarchy is necessarily a matter of subjective choice, and the questions of how to create a hierarchy and what models are desirable in a canonical hierarchy has generated extensive discussions (Jeevanjee et al., 2017). Our purpose here is to present a modelling framework which enables climate scientists to easily and transparently traverse the specific model hierarchy suiting their needs.

Designing and building a framework that facilitates traversing this hierarchy remains a challenge. The lack of flexibility of existing climate models forces scientists to spend a lot of time reading and modifying code to construct alternative model versions that should in principle be straightforward to build. Most modelling frameworks simply provide code to exchange information between different physical domains such as atmosphere and ocean (See Theurich et al., 2015; Valcke et al., 2012, for a survey of modelling frameworks), with each physical domain being represented by a monolithic block of code. It was

only with the advent a new generation of frameworks like the Earth System Modelling Framework (ESMF) (DeLuca et al., 2012; Theurich et al., 2015) that fine-grained control over the components that constitute a climate model was made possible. For instance, ESMF allows configuring components as trees, where the leaf nodes could represent physical processes such as radiation and the root of the tree could represent a physical domain such as the atmosphere. Such a hierarchical ordering of physical processes is also present in Python based modelling packages – previous versions of `climt` and `climlab` (https:

15    //github.com/brian-rose/climlab) allow building models in a manner similar to ESMF. However, the norm continues to be that climate model composition is configured by namelist variables and boolean flags in the code rather than framework-based approaches (like the component trees that ESMF allows).

       Another emerging concern in the scientific community is that of reproducibility of research (Peng, 2011). While publicly available climate models do provide validated configurations that are in principle completely reproducible, climate scientists

routinely need to make changes to the code that are hard to document (or understand). Short of sharing a copy of the entire code base, such modifications makes it difficult to reproduce simulations. While some level of code manipulation is inescapable, we note from our own experience and from reading about such modifications in the literature that most of them follow set patterns which could easily be provided by modelling frameworks themselves.

       In this paper we present a new modelling framework, `sympl`, and a model toolkit, `climt`. While `sympl` focuses on

providing a model framework, a rich taxonomy of components, and model agnostic configuration options, `climt` focuses on providing a broad array of physical components to allow users to build scientifically useful models. `climt` also provides model dependent configuration options and helper functions to create an initial model state as required by the components.

       `sympl` and `climt` allow finer grained control over the composition of the model, with individual components representing physical processes (such as radiation, convection, etc.,) rather than physical domains. Attempting to model the climate system

at the physical process level has its own challenges which we attempt to solve in these packages. Initiatives to build frameworks to traverse the hierarchy between highly idealised models to full scale GCMs (general circulation models) do exist (Fraedrich et al., 2005; Vallis et al., 2017), but we believe the definition of a clear set of classes to represent the physical process level of the model to be unique.

       `sympl` and `climt` allow writing models which are easy to use and facilitate reproducibility of simulations. Both these

packages are subject to ongoing development, but have reached a level of maturity that makes it worthwhile to document them

here. In section 2, present a series of models written using `sympl` and `climt` which illustrate the construction of a model hierarchy. In section 3 we describe some challenges that modelling frameworks have to solve (in the context of the above examples when possible) and discuss how `sympl` and `climt` address these challenges. We then discuss in more detail the interfaces of `sympl` (section 5) and `climt` (section 6). In section 7 we present some benchmark calculations, and conclude with a discussion of developments planned in the future.

## 2  `sympl` and `climt` in action

To illustrate the advantages of the fine-grained control that `sympl` and `climt` offer, we consider a series of examples starting with a diagnostic radiative calculation and ending with an idealised three dimensional atmospheric general circulation model.

Figure 1 shows the script required to calculate the heating tendencies from a longwave radiative transfer component (Clough et al., 2005). A detailed explanation of the script follows:

- **Line 1**: Import the `climt` package.

- **Line 4**: Instantiate a radiative transfer component.

- **Line 7**: Create a state dictionary which contains all the quantities required as inputs by the radiative transfer component.

- **Line 9**: Calculate the heating rate (available in `tendencies` and any associated diagnostics such as the radiative fluxes (available in `diagnostics`).

In Figure 1, we have used the default values for quantities in `model_state`, which corresponds to an isothermal atmosphere.

This example, though seemingly simple, is remarkable because of the ease with which such a diagnostic calculation can be performed. Traditionally, such a calculation would involve writing a Fortran "driver", compiling it with the radiative transfer library, writing the output to a file, and then reading the output file into a suitable environment for further analysis. The ease of use illustrated in the above example is a direct result of the fine-grained control that `sympl` and `climt` allow – individual components can be configured and run (interactively, if desired) without having to compile them with a driver file. To summarise, *components*, not models, are first class entities within the `sympl` framework.

It is worth examining this example in greater detail, since it highlights some important features of `sympl` and `climt` that we will look at closely in subsequent sections. The component called `radiation` is an implementation of the `sympl` entity called `TendencyComponent`, which is a template for components which calculate tendencies of a quantity (air temperature in this case) based on quantities in a state dictionary. We will encounter other kinds of components in subsequent examples.

Figure 2 builds upon the previous example to create a model that includes both radiation and convection, steps the state quantities forward in time, and writes the output to a file. The changes from the previous script are as follows:

- **Line 7**: Define the model time step using `timedelta` from the `datetime` library, which is part of any Python distribution.

```
import climt
# Create component
radiation = climt.RRTMGLongwave()
# Create model state
model_state = climt.get_default_state([radiation])
tendencies, diagnostics = radiation(model_state)
```

**Figure 1.** A Python script which calculates the heating tendencies and associated diagnostics from a longwave radiative transfer component. See the text for a detailed description.

- **Line 21**: Create a time integrator which steps the model state forward in time, using tendencies generated by the radiation and convection components. The integrator chosen is an instance of the `sympl` `Stepper` class which implements a variety of Adams-Bashforth schemes.

- **Line 24**: Create a "monitor" component which writes the model state to a netCDF file.

- **Lines 27-37**: The boundary layer component provides new values of model quantities, which are used to update the model state. The time integrator incorporates the tendencies due to radiation and convection and provides new values for the model state as well. The current model state is updated with diagnostics and written to a netCDF file. The model state is then updated with the new model quantities to prepare for the next iteration.

This example illustrates how to piece together a radiative-convective equilibrium (RCE) model from different kinds of
`sympl` components. This example also moves up the model hierarchy, away from static diagnostic calculations to a model which evolves in time. It is worth noting that the first half of the example which creates components and a model state remains identical to the procedure followed in the previous example. This example shows two notable features of the design of `sympl` and `climt` – One, individual components can step the model state forward themselves and Two, the model integrator is a separate entity in itself which can be replaced easily (to use more stable integration schemes, for example). These features are
in keeping with our goal of capturing the diversity of model components – some of which produce tendencies, and others that produce new state quantities – and allowing the user fine-grained control over model configuration.

This example also illustrates how `sympl`'s design provides a clear understanding of the model to users. Configuring the model consists of modifying a run script which is meant to be entirely legible to the user. By reading the run script, one can see exactly which model components are being used, how the state is initialised, what configuration options are being passed
to which components, what time integration scheme is used on which components, the order in which components are being called, and the point within the integration where the state is being output to a file.

```
from sympl import (
AdamsBashforth, NetCDFMonitor)
import climt
from datetime import timedelta
# Define model timestep in minutes
model_timestep = timedelta(minutes=1)
# Create components
radiation = climt.RRTMGLongwave()
convection = climt.EmanuelConvection()
boundary_layer = climt.SimplePhysics()
# Create model state
model_state = climt.get_default_state(
[radiation, convection, boundary_layer])
# Create integrator
time_stepper = AdamsBashforth(
[radiation, convection])
# Create monitor
monitor = NetCDFMonitor('radiative_convective.nc')
# step model forward
for step in range(10):
bl_diagnostics, bl_new_state = boundary_layer(
model_state, model_timestep)
model_state.update(bl_diagnostics)
model_state.update(bl_new_state)
diagnostics, new_state = time_stepper(
model_state, model_timestep)
model_state.update(diagnostics)
monitor.store(model_state)
model_state.update(new_state)
model_state['time'] += model_timestep
```

**Figure 2.** A Python script which calculates the radiative-convective equilibrium temperature of an atmospheric column for a fixed surface temperature. See the text for a detailed description.

```python
from sympl import (
TimeDifferencingWrapper, NetCDFMonitor)
import climt
from datetime import timedelta
# Define model timestep in minutes
model_timestep = timedelta(minutes=1)
# Create Components
radiation = climt.RRTMGLongwave()
convection = climt.EmanuelConvection()
boundary_layer = TimeDifferencingWrapper(
climt.SimplePhysics())
time_stepper = GFSDynamicalCore(
[radiation, convection, boundary_layer])
# Create model grid
model_grid = climt.get_grid(nx=64, ny=64, nz=28)
# Create model state
model_state = climt.get_default_state(
[time_stepper], grid_state=model_grid)
# Create monitor
monitor = NetCDFMonitor('moist_agcm.nc')
# step model forward
for step in range(10):
diagnostics, new_state = time_stepper(
model_state, model_timestep)
model_state.update(diagnostics)
monitor.store(model_state)
model_state.update(new_state)
model_state['time'] += model_timestep
```

**Figure 3.** A Python script which creates an idealised moist atmospheric general circulation model. See text for a detailed description.

Figure 3 presents the next step in the hierarchy from Fig. 2, creating a moist atmospheric general circulation model (AGCM) in an aqua-planet configuration with a prescribed sea surface temperature. The default surface temperature is horizontally uniform, but we omit prescribing a realistic temperature distribution for purposes of comparison with Fig. 2. The main differences from the previous example are:

– **Line 14**: The boundary layer component is wrapped to output tendencies instead of a new state. This wrapper is required since the spectral dynamical core must apply tendencies to most quantities in spectral space for numerical reasons.

    – **Line 18**: The spectral dynamical core is used as the time stepper instead of the Adams-Bashforth scheme used previously.

    – **Lines 22 and 25**: Since the model now has three dimensions, we require a grid describing the latitudes, longitudes as well. Line 18 creates a model grid, and the default model state created in Line 21 uses this grid to create an appropriate

three dimensional state.

A remarkable fact about this example is that it is only one line longer than the previous example. Intuitively, this seems appropriate – an AGCM can be thought of as a collection of radiative-convective columns which communicate with each other via the dynamical core. However, in most modelling frameworks the intuitive picture of the transition from an RCE to an AGCM does not easily translate to code. This plug-and-play behaviour is a direct consequence of the modularity of individual

`sympl` components and the fact that all components of a similar kind (`AdamsBashforth` and `GFSDynamicalCore` in this case) implement the same interface, making it possible to reuse almost the entire model script from the RCE case. In this way, `sympl` and `climt` allow constructing models in such a way that a change that intuitively seems small also translates to a code change that is small.

In this following section, we look at detail at the design decisions that allow for the construction of a model hierarchy as

described in the three examples in this section.

## 3   Design considerations and choices

In this paper we distinguish between modelling frameworks, model toolkits, and models themselves. A framework (such as `sympl`) consists of abstractions of "infrastructure" code that allow the creation of climate models. A framework creates rules one must follow, but by doing so ensures models using the framework are easier to write, understand, and combine with one

another. A toolkit (such as `climt`) implements those abstractions for concrete physical processes, and may provide additional functionality that is not covered by the framework. A model itself (such as in Figures 2 and 3) is written using components that may come from a toolkit which follows the guidelines of the framework.

Modelling frameworks should enable scientists to intuitively combine model components and create an appropriately complex model for the scientific question at hand. The user should also be able to specify details such as the order in which

components are called and the time stepping schemes used. Model toolkits should provide a wide variety of components that enable users to write a model appropriate to the question at hand. Toolkits should also maintain a list of quantities and numerical grids that are required by the components it provides to facilitate creating model arrays. It is also desirable that the process

of creating a model is fairly easy to understand and that the model code be self-documenting to eliminate the need to write additional documentation whenever possible.

## 3.1 Component diversity

One of the major aims of `sympl` and `climt` is to allow fine-grained control over the processes that constitute an Earth system model. In currently available modelling frameworks, the integration of scientific code (or model components) and the modelling framework happens at the physical domain level – atmosphere, ocean, land and so on. The processes that operate within each physical domain (fluid dynamics, radiation, convection etc.,) are not accessible in a systematic manner, and their code becomes tightly coupled and difficult to modify. For example, changing the radiation code in an atmospheric model is typically harder than changing the kind of ocean the atmosphere is coupled to (slab, dynamical, etc.,). Tight coupling of code at the process level can make these models more efficient, but at the cost of scientist efficiency, as the code is significantly less flexible and have undocumented and complicated interfaces between components.

Furthermore, the fact that certain process level components are written to work only with certain other components demands an "architectural unity" (Randall, 1996) which might also encourage tight integration of model components. Within `sympl`, it may still be the case that two components are incompatible with one another (for example, using different thermodynamic quantities), but because their interfaces are clearly defined, it is easier to couple these components (for example, by converting thermodynamic quantities using an additional component). Other incompatibilities are handled automatically. For example, `sympl` automatically ensures components which use different units or dimension orderings can be used alongside one another.

If we allow for configuration at the process level, we are then faced with model components which behave quite differently: Some components (like radiation) return tendencies, while others (like large scale condensation) return a new value of a physical quantity. `sympl` provides a set of component classes that is comprehensive enough to capture the diversity of component behaviours.

An example of a `sympl` component is depicted in Fig. 4. `input_properties` and `tendency_properties` are 'property' attributes containing details of the required inputs and returned outputs, tendencies or diagnostics. Here, the input is a quantity `air_temperature` whose horizontal dimensions are `latitude`, `longitude` and whose values in the vertical are defined at model mid-levels. The units are specified for each quantity. The `array_call` method will be called by `sympl` with numpy arrays that are automatically extracted from the model state to satisfy the input property specifications. It is written to return numpy arrays as output which satisfy its tendency and diagnostic property specifications. The `__call__` method which is called, for example, in Line 9 of Fig. 1, is implemented by the base `sympl` classes. This method encapsulates the boilerplate code of performing consistency checks on the model state, extracting numpy arrays with the correct units and shape, and creating a new state dictionary from the arrays returned by `array_call`.

Dimension and unit requirements in `sympl` are not restrictions on the inputs, but rather describe the internal representation used by the component. `sympl` will automatically convert the input state to satisfy these requirements, and raise an exception if that is not possible. In this way, the property dictionaries act as self-documenting code, which both documents the component interface and is used to convert input arrays to the desired dimension ordering and units for the component.

```python
lass PrescribedHeating(TendencyComponent):
input_properties = {
'latitude': {
'dims': ['*'],
'units': 'degrees_N',
},
'longitude': {
'dims': ['*'],
'units': 'degrees_E',
},
'air_pressure': {
'dims': ['mid_levels', '*'],
'units': 'Pa',
},
}
diagnostic_properties = {}
tendency_properties = {
'air_temperature': {
'dims': ['mid_levels', '*'],
'units': 'degK s^-1',
}
}
def __init__(self, forcing_filename, **kwargs):
[...]
super(PrescribedHeating, self).__init__(**kwargs)
def array_call(self, state):
[...]
```

**Figure 4.** The general code layout for a `sympl` component.

As a `TendencyComponent` (discussed in Section 5), this component outputs tendencies of a quantity `air_temperature`. The `array_call()` method accepts a dictionary containing just the `numpy` arrays extracted from the model state with the correct units and dimensions and returns the temperature tendency as specified in the component properties.

Since components in `sympl` and `climt` are first class entities, they are not dependent on any other code for execution –
5 Fig. 1 shows how to interact solely with a radiative component without the need for an integrator or any other entity. This behaviour serves an important educational purpose and facilitates diagnostic calculations made very often during research.

## 3.2 Model configuration

Together, `sympl` and `climt` allow a natural way of configuring various aspects of a climate model listed below:

- Physical configuration (toolkit agnostic): the physical constants required by the model components.

- Algorithmic configuration (toolkit specific): the "tunable" parameters which modify the behaviour of the algorithm which represent physical processes – for example, the entrainment coefficient in a convective parameterisation.

- Interfacial configuration (toolkit agnostic): Modifications applied to inputs and outputs at the interface of a component, further described below.

- Memory and Computing resource configuration (toolkit specific): The layout of arrays used by the model and the distribution of the model components over the available number of processors and co-processors.

- Compositional configuration (toolkit specific): The components that compose a model, any dependency between components, the order in which components are executed all need to be described.

Figure 5 shows types of configuration information and where such configuration lies `climt`/`sympl` and in traditional climate models. In contrast to `sympl` where all configuration passes through a readable run script, the sheer variety of locations where the configuration resides in traditional models makes it hard to keep track of what configuration information has changed, and how configuration changes affect model runs. This makes it daunting for beginners to write models. Model configuration using `sympl` and `climt` is highly centralised and easily accessible - all configuration passes through the model run script, which is written to be readable and accessible to model users. Such centralised configuration reduces errors arising due to misconfiguration of the model.

While most of the configuration elements listed above are familiar, interfacial configuration is mostly unheard of and usually applied in a non-systematic manner within climate models. The `TimeDifferencingWrapper` used in line 14 of Figure 3 is an example of interfacial configuration of a climate model component. There are a variety of interfacial modifications which are commonly applied to components, and can be applied in a consistent manner across components. For instance, model components could

- Normally return a new value rather than a tendency of some physical quantity, but in certain instances a tendency might be desirable (as in Figure 3).

- Provide output that is piece-wise constant in time – the output is updated only once every $N$ iterations, and the same value it output for the next $N - 1$ iterations. This behaviour is normally used in radiative transfer codes.

- Scale some of its inputs or outputs by some floating point number. This kind of behaviour is desirable for example in mechanism denial studies.

We can interfacially modify certain behaviours of model components, such as the ones above, by interacting with only the inputs and outputs of scientific components. `sympl` formalises such configuration by providing wrapper objects like `TimeDifferencingWrapper`.

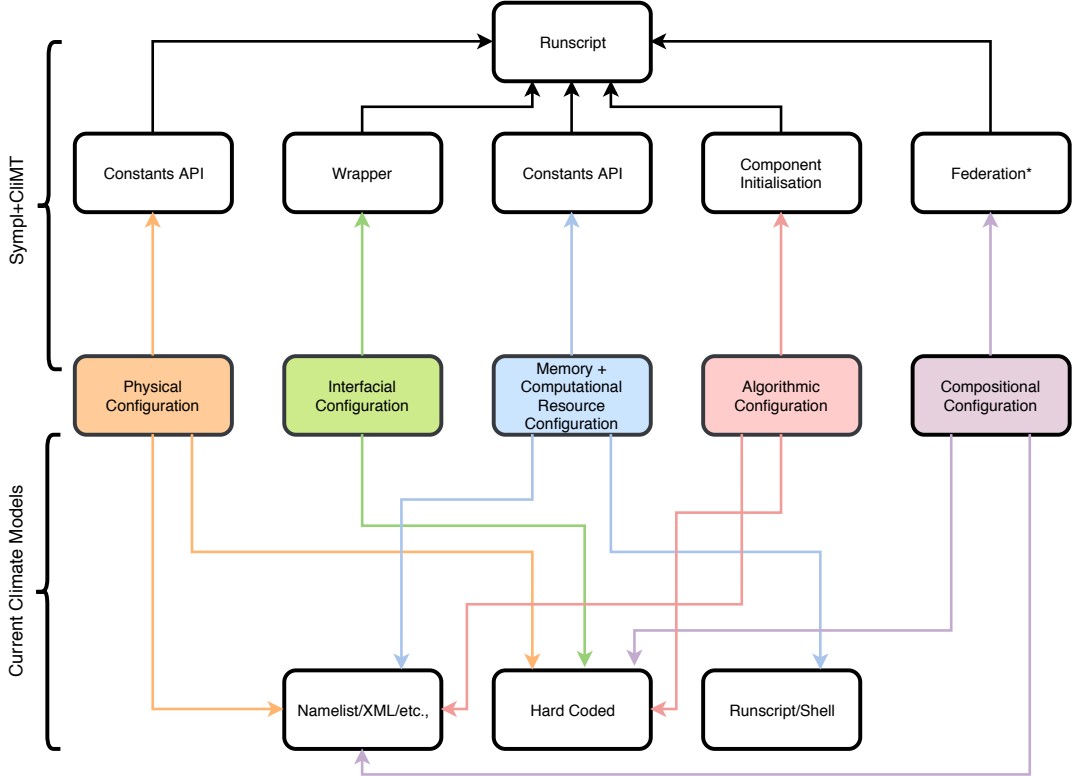

**Figure 5.** The variety of configuration options in a climate model and in the `sympl-climt` framework. The coloured boxes indicate the type of configuration and the white boxes parts of the model code base. Arrows from the coloured to the white boxes indicate the part of the code base that is typically responsible for the configuration from which the arrow originates. A particular type of configuration could exist in two different parts of the code base: such a situation is indicated by multiple arrows terminating at all the relevant white boxes. Note that not every climate model uses all configuration options. The starred box on the `sympl+climt` side indicates functionality that is not yet implemented.

## 4 Model arrays

Model arrays are contained within the `DataArray` abstraction provided by `xarray` (Hoyer and Hamman, 2017) rather than the more low-level `numpy` array. `DataArrays` are an abstraction over `numpy` arrays with a more natural fit to climate data by providing labeled dimensions and metadata storage capabilities.

5    `sympl`'s `DataArray` object is a subclass of the `xarray` `DataArray` that provides units handling and conversion and will be described subsequently. This exposes the powerful analysis capabilities of `xarray`, allowing users to build an end-to-end pipeline entirely within Python, from simulation to data analysis and generation of publication-ready figures.

Since low-level array operations using `numpy` and `xarray` are fairly simple, especially changing coordinate ordering and C/Fortran memory ordering, `climt` only provides guidelines for memory layout of arrays. However, changing the ordering of dimensions in memory will incur the performance penalty of copying the array if the model calls compiled code.

In the interest of readability of component and model code, `sympl` strongly encourages and `climt` makes use of descriptive names for model quantities, adhering to the CF conventions[1] when possible. Most pre-defined quantities in `climt` use names derived from the CF conventions. One additional suffix that we found necessary to use was `on_interface_levels` to distinguish between quantities defined on the interfaces and mid-levels of the vertical grid. For example `air_temperature` refers to the air temperature defined at the vertical grid centre, whereas `air_temperature_on_interface_levels` refers to the same quantity defined at the vertical grid edges. This convention is only a requirement within the model state. Within an individual component, shorter names can be used for variables representing quantities. This shorter name is also contained in the property dictionary of the component, serving as documentation for the meaning of that shorter variable name.

## 4.1 Modelling language

Python was used as the language to write the framework. Python as a language and the Python ecosystem have a number of desirable features, all of which were taken advantage of during the development of `sympl` and `climt`:

- Earlier versions of `climt` were written in Python, which gave the authors an idea of the convenience and flexibility it afforded. In particular, the object-oriented capabilities of Python provide a straightforward way to represent the component based architecture adopted by almost all climate modelling frameworks.

- Scientific libraries within the Python ecosystem now offer acceptable performance for computationally intensive operations typically used in climate models.

- The Python ecosystem includes many libraries which can be useful in developing climate models. Examples include machine learning, graphics and web services libraries.

- Python's ability to act as a glue language allows interfacing with the large number of libraries for climate modelling already available in Fortran.

- Tools available in the Python ecosystem like `jupyter`, `pytest` and `sphinx` which enable writing reproducible workflows and code that is well documented and tested.

## 5 `sympl` – Design and programming interface

`sympl` conceives of a climate model as a state that is continuously updated by various components. `sympl`'s taxonomy consists of seven kinds of components. Four of these component types are used to represent physical processes and the remaining represent other functionality required to build and run models.

---

[1] http://cfconventions.org/Data/cf-standard-names/48/build/cf-standard-name-table.html

- TendencyComponent objects like RRTMGLongwave in Fig. 1 which take the model state as input and returns tendencies of quantities and values of quantities defined at the time of the input state.

- Stepper components like SimplePhysics in Fig. 2 which take the model state and a timestep as input and returns values of quantities defined at a new time (after the timestep) and optionally diagnostic values of quantities defined at the time of the input state. These are implicit as they define the target model state in terms of the target model state (e.g. that the target state is not supersaturated).

- DiagnosticComponent objects which take the model state as input and return quantities defined at the time of the input state as output.

- ImplicitTendencyComponent objects like EmanuelConvection in Fig. 2 which return tendencies, but require the model timestep to produce these tendencies. This is required when the tendencies are defined in terms of the target model state, as is often done in convection schemes or flux limiters. These should generally be written as Stepper components which can later be wrapped into ImplicitTendencyComponent objects if needed (such as with SimplePhysics in line 14 of Fig. 3).

- TendencyStepper components like AdamsBashforth in Fig. 2 which contain a set of TendencyComponent objects and use the tendencies they output to integrate the model state forward in time.

- Monitor components like NetCDFMonitor in Fig. 2 which provide a store method which takes the model state as input and "stores" it. The implementation of this method is left to the user, and currently is used for NetCDF output and plotting.

- Wrapper components like TimeDifferencingWrapper in Fig. 3 which contain other sympl components and modify the inputs passed to or outputs generated by the 'wrapped' component.

Schematics of how the above components interact with the model state are presented in Figure 6. A DiagnosticComponent object (Panel a) is very simple, producing diagnostic quantities that are inserted or updated in the current model state. Monitor objects (Panel b) take in the model state and perform some action using it. Slightly more complicated is the Stepper object (Panel c), which steps the model state forward in time. In addition to producing a new state, it can produce diagnostic quantities which are inserted or updated in the current model state. Panel d is the most complex, depicting how TendencyComponent objects are used with a TendencyStepper to update the model state. Once created, a TendencyStepper behaves exactly like an Stepper object. Internally, it provides the input state to the TendencyComponent objects to compute tendencies, and uses those tendencies according to its time stepping scheme to evolve the model state forward in time. The TendencyStepper provides the same model state to all TendencyComponent objects it contains and sums the tendencies before stepping forward in time (See Fig. 6d). Using other time marching algorithms such as sequential tendency or sequential update splitting (Donahue and Caldwell, 2018) will require users to implement their own TendencyStepper object, or to call several Stepper and TendencyStepper components in sequence.

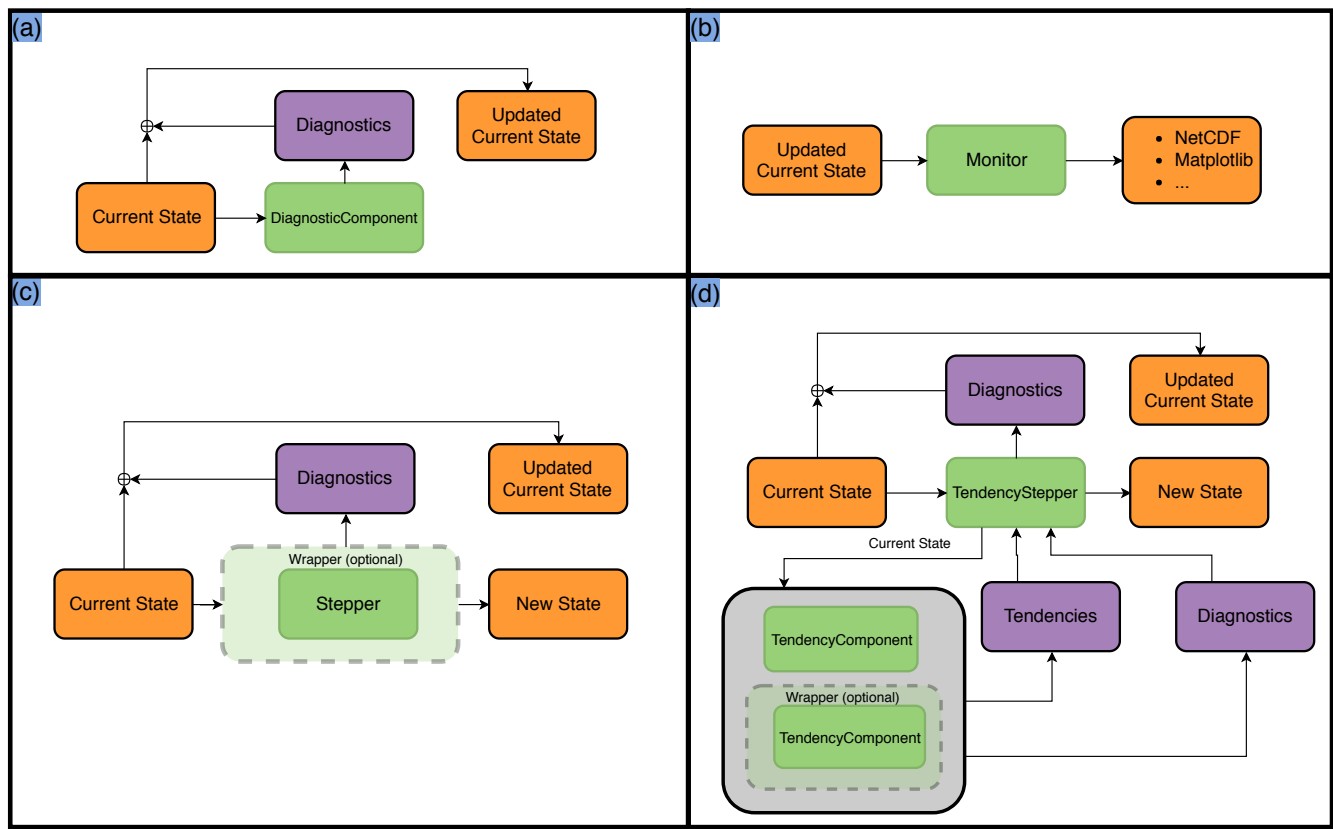

**Figure 6.** Flow of data for each type of component in `sympl`. The four panels show: **a**) `DiagnosticComponent`, which creates diagnostics (purple box) based on the current state. The current state is updated with the resulting diagnostics, resulting in the updated current state. **b**) `Monitor`, which can "store" the updated current state into some format like NetCDF or plots. **c**) `Stepper`, which determines a new state from the current state and a time step, and also diagnostics from the time of the current state which are used to update the current state. The current state could then be passed on to `Monitor` components (see panel **b**). **d**) `TendencyStepper`, which is a special case of `Stepper` initialised with a list of `TendencyComponent` objects (denoted by green boxes within a grey box). It passes the current state on to those `TendencyComponent` objects to compute tendencies and diagnostics, which are used to compute its outputs (generally using a time stepping scheme). In all figures, dark green boxes denote components, light green boxes denote optional wrappers, orange boxes indicate the state dictionary at different times and purple boxes indicate tendencies and diagnostics generated by components. The converging arrows at a summation symbol (plus sign inscribed within a circle) denotes updating the state dictionary (orange) to include output (purple) quantities. Examples of wrapper placements are not exhaustive and meant only as examples.

As mentioned previously, wrapper components contain a "wrapped" component and modify the inputs or outputs, changing how the component appears to behave. Currently, `sympl` has the following wrappers:

- `TimeDifferencingWrapper` creates tendencies from the output of an `Stepper` component by first order differencing. This creates an `ImplicitTendencyComponent` from an `Stepper` component, which is required when using spectral methods.

- `UpdateFrequencyWrapper` calls a wrapped `TendencyComponent` only after the user-specified time interval has elapsed, and until then outputs the previously returned value. In effect, this creates a piecewise constant output tendency which can reduce the computational load during a simulation. This is often used on radiation schemes.

- `ScalingWrapper` scales the inputs passed into the wrapped component and the outputs (new state, tendency or diagnostic) returned by the component.

This taxonomy of components is larger than those typically used in modelling frameworks. For example, ESMF only considers two kinds of components – Gridded and Coupler components. However, as discussed previously, this extended taxonomy is required to capture the diversity of components that arise if models are written to be configurable and modular at the process level.

## 5.1 Model State and the `DataArray` abstraction

The model state is a dictionary whose keys are the names of model quantities and values are `sympl` `DataArray` objects. The model state also contains a required keyword `time` whose value is an object that implements the Python `datetime` or `timedelta` interface. `sympl` provides an interface to use the `datetime` objects from the `cftime`[2] package to support several different calendars, as well as dates not supported by the `numpy` `datetime64` or built-in `datetime` objects. A schematic of the model state is presented in Fig. 7. `sympl` does not put hard restrictions on the name of model quantities, though standardised names should be used to ensure inter-package compatibility. All `DataArray` objects must define a string attribute called `units`, which is used to convert the data contained within to the appropriate units requested by a component. The units conversion is performed internally using the `Pint`[3] library. Since the actual contents of the state is dependent on model details, `sympl` assumes that the initialisation of the model state will be done by a model package (such as `climt`, or by the user.

## 5.2 Physical Constants

`sympl` maintains a unit-aware library of constants which can be accessed or modified by model packages and by the user through `get_constant()` and `set_constant()` functions. For example, `planetary_rotation_rate` can be changed with a single function call. The unit handling is important to ensure constants are given to components in the units

---

[2]https://github.com/Unidata/cftime
[3]https://pint.readthedocs.io/en/latest/

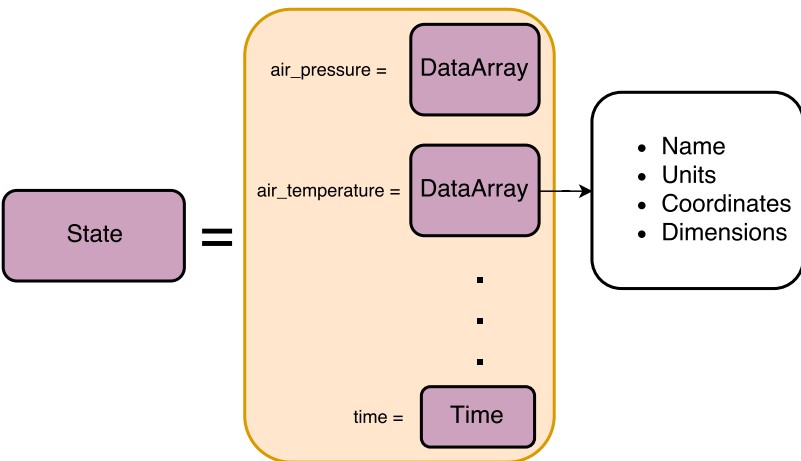

**Figure 7.** The model state as in the `sympl` framework. The state in the orange box contains all the information that is stepped forward in time. Each `DataArray` contains information such as the quantity name, units and dimensions/coordinates.

they each require. For example, the RRTMG radiative transfer code (Clough et al., 2005), requires physical constants in CGS units.

For the purposes of logging, physical constants are classified into various categories:

   – planetary constants such as rotation rate, acceleration due to gravity.

5   – physical constants such as the speed of light.

   – atmospheric constants such as specific heat of dry air and reference air pressure.

   – stellar constants such as stellar irradiance

   – condensible constants which refer to the thermodynamic properties of the condensible (in all three phases) in the atmosphere.

10   – Oceanographic constants such as the reference sea water density.

We chose to keep the constants related to the condensible component of the atmosphere separate to ensure `sympl` is flexible enough to handle general planetary atmospheres. `sympl` provides a function `set_condensible()` which allows switching all constants related to the condensible. For example `set_condensible('methane')` will replace all condensible constants (such as density of liquid/solid/gaseous phases, latent heat of condensation) to those corresponding to methane, provided 15 such constants are already in the constants dictionary. The default condensible is water, which is currently the only condensible compound for which default values are given by `sympl`.

### 5.3 Modelling using `sympl`

A typical workflow when using a model written using `sympl`, as seen in previous examples such as Fig. 2 and Fig. 3, might involve the following steps:

1. Initialise model components, providing configuration information.

2. Use `Wrapper` components to modify the behaviour of any components if necessary.

3. Intialise model state which contains all quantities required by the selected components.

4. Use `TendencyStepper` to collect all `TendencyComponent` components into a component that can step the model state forward in time.

5. Begin the model main loop.

6. Call `DiagnosticComponent` to compute any derived quantities from prognostic quantities or provide forcing quantities at a given time step.

7. Call `Stepper` components and get a new state dictionary with the updated model quantities and any diagnostics. Update the initial model state with diagnostics.

8. Call `TendencyStepper` and get a new state dictionary with the updated model quantities and any diagnostics. Update the initial model state with diagnostics.

9. Call any `Monitor` components to store the initial model state (e.g. store to disk, display in real time, send over the network).

10. Increment model time and repeat the model main loop.

## 6  `climt` – Design and programming interface

### 6.1  Model state, quantity dimensions and output dictionaries

For initialisation, `climt` provides the functions `get_grid()` and `get_default_state()`. `get_grid()` creates quantities that define the grid such as latitude, longitude, and air pressure. `get_default_state()` accepts a list of components and optionally a state with grid quantities and creates a state dictionary which satisfies the input requirements for those components. Default values of each model quantity are defined centrally in `climt`. The default values provided are scientifically meaningful, and can be used without modification for certain simulations.

## 6.2 Model Composition

Currently, the creation of the model and running the simulation loop is done by hand, which provides better understanding of what the model is doing but increases the verbosity of model code. In the near future, `climt` will provide an additional class called `Federation` which automates the process of creating a model from its components. `Federation` would not require that the user know the difference between a `TendencyComponent` and `Stepper` (for instance) and their different call signatures, or that `TendencyComponents` require a `TendencyStepper` to step the model state forward in time. This makes creating models easy especially for those who are not familiar with climate modelling. The tradeoff is that the run script will not explicitly describe the sequence of the main loop because that information is hidden within the `Federation` code, but this can be desirable for certain applications, particularly in education. As mentioned before, this automation is possible only because of the rich taxonomy of components `sympl` provides.

## 6.3 Features and software engineering

`climt` currently has the following components that can be used to build models[4]:

– RRTMG longwave and shortwave radiative transfer (Clough et al., 2005): This is a Fortran component accessed via a cython wrapper. RRTMG is a state-of-the-art radiative transfer code used in many climate models.

– Grey gas radiation scheme: simulates radiative transfer in a grey gas. This component is accompanied by another component that provides an optical depth distribution which mimics the effect of water vapour (Frierson et al., 2006). These components are written in pure Python. This radiative scheme has been used in many idealised climate dynamics simulations to isolate the thermodynamic effects of latent heat release from the radiative effects of water vapour, which is a strong greenhouse gas.

– Insolation: This component is written in pure Python. It provides the solar zenith angle based on the time available in the model state. This zenith angle is used in radiative transfer codes. Currently, this component uses approximations and orbital parameters which make it highly accurate for earth but inapplicable to other planets.

– Emanuel convection scheme (Emanuel and Živković Rothman, 1999): This is a Fortran component accessed via a Cython wrapper. It is a mass flux based convection scheme which is based on the boundary layer quasi-equilibrium hypothesis (Raymond, 1995).

– Grid scale condensation: This is written in pure Python. It calculates the water vapour and temperature fields in the atmosphere after condensing out excess water vapour to keep the atmospheric column from becoming super-saturated.

– Spectral dynamical core: This component is derived from the General Forecast System (https://github.com/jswhit/gfs-dycore). It is a Fortran module accessed via a cython wrapper. It uses a high performance spherical harmonics library `shtns`

---

[4]All components written in pure Python were written by the authors for use in `climt`.

(https://bitbucket.org/nschaeff/shtns). It is parallelised using OpenMP (Dagum and Menon, 1998), and therefore is most effective on shared memory systems. The dynamics is stepped using an implicit-explicit total variation diminishing Runge-Kutta 3 time-stepper. The physics tendencies are stepped forward using a forward Euler scheme.

- Simple Physics package for idealised simulations (Reed and Jablonowski, 2012): This is a fortran module accessed via a cython wrapper. It provides initial conditions which can be used for testing moist dynamical cores, and also provides a simple diffusive boundary layer suitable for idealised simulations.

- Slab surface: This component is written in pure Python. It allows for a prognostic surface temperature by calculating the surface energy budget. It is flexible enough to represent land or ocean. It currently does not account for localised heat fluxes.

- Sea/land ice model which allows for snow and ice layers, and energy balanced top and bottom surfaces. This component is written in pure Python. It is flexible enough to represent ice/snow growth and melting. It is capable of representing sea or land ice based on the surface type available in the model state. It currently cannot handle fractional land surface types.

- Held-Suarez forcing (Held and Suarez, 1994): This component is written in pure Python. It provides an idealised set of model physics which can be used for testing dry dynamical cores and idealised simulations.

- Initial conditions from the dynamical core MIP (DCMIP) (https://www.earthsystemcog.org/projects/dcmip/): This is a Fortran module accessed via a cython wrapper. It provides initial conditions for a wide variety of tests which allow assessing the conservation properties of dynamical cores.

This set of components allow building a hierarchy of models ranging from single column radiative-convective models to energy balanced moist atmospheric general circulation models. Because of the fine-grained configurability of `sympl`/`climt`, the difference between the number of lines of code required to build a single column model and a moist GCM is only around 10 lines of Python code (see scripts in supplementary material). More importantly, most of the code is reusable when moving from a simpler to a more complex model.

Both `sympl` and `climt` are open source projects, licensed under a permissive BSD license. Both packages are available on Mac, Linux and Windows platforms, and can be directly installed from the Python Package Index using one line commands:

```
pip install sympl
pip install climt
```

eliminating the need to download source code from GitHub. The Python Package Index projects are located at https://pypi.python.org/pypi/sympl and https://pypi.python.org/pypi/climt respectively.

`climt` also provides binary releases on all supported platforms, eliminating the need to have a compiler on the user's system. `sympl` is written in pure Python, and does not have any compiler requirements. Both packages are regression tested using the online services TravisCI (https://travis-ci.org/) and AppVeyor (https://www.appveyor.com/). Both packages also maintain regularly updated documentation at http://sympl.readthedocs.io/en/latest/ and http://climt.readthedocs.io/en/latest/.

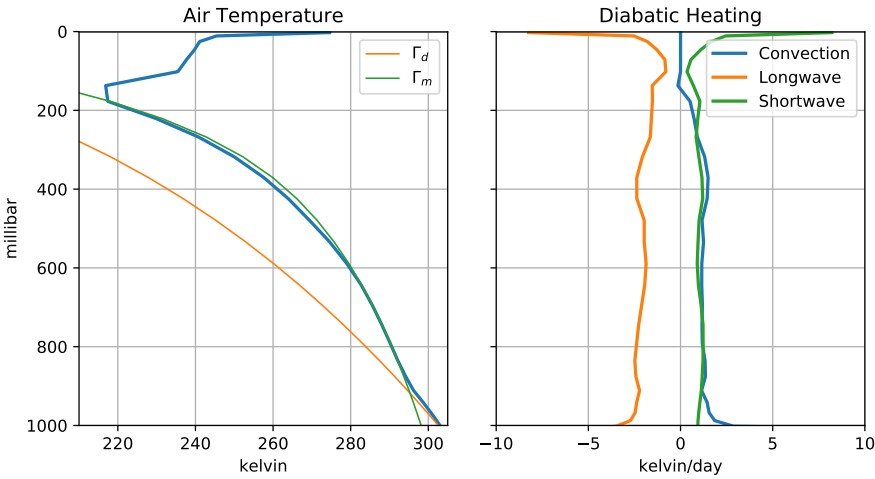

**Figure 8.** The mean equilibrium profiles in the radiative convective single column model. The mean temperature is presented in the left panel, along with the dry and moist adiabats which are orange and green respectively. The mean heating profiles are presented in the right panel, which shows a stratosphere in radiative equilibrium and a troposphere in radiative-convective equilibrium.

## 7   Some benchmark simulations

The first simulation is that of an atmospheric column that is run to equilibrium in the presence of radiation and convection. This model uses the RRTMG longwave and shortwave components, the Emanuel convection scheme, the Simple Physics component as its boundary layer scheme and a slab ocean of thickness 50 metres. The model timestep is 5 minutes and the results presented in Fig. 8 are the mean between 15000 and 20000 timesteps. The air temperature transitions from a dry adiabat in the boundary layer to a moist adiabat in the free atmosphere until the tropopause. The diabatic heating balance changes from a balance between radiation and convection in the troposphere of the model to a pure radiative equilibrium in the stratosphere.

The second simulation is of a idealised aquaplanet GCM with fixed equinoctial insolation. As mentioned before, the modular nature of our framework allows re-use of much of the runscript code from the above single column model. It consists of all the components used in the previous model along with a dynamical core which is used as the time stepper. The model was run for two years and the results presented are the mean over the last six months. The simulated climate of the model is as expected from such a configuration: the zonal mean zonal winds show two strong westerly jets which penetrate to the surface. The zonal mean temperature shows a distinct tropical cold point and an increase in the temperature above the tropopause. The zonal mean convective heating rate shows deep heating in the tropics and much shallower heating in the subtropical areas dominated by descent of air. This simulation ran at a resolution of 128 longitudes, 62 latitudes (or T42 resolution) and 28 levels.

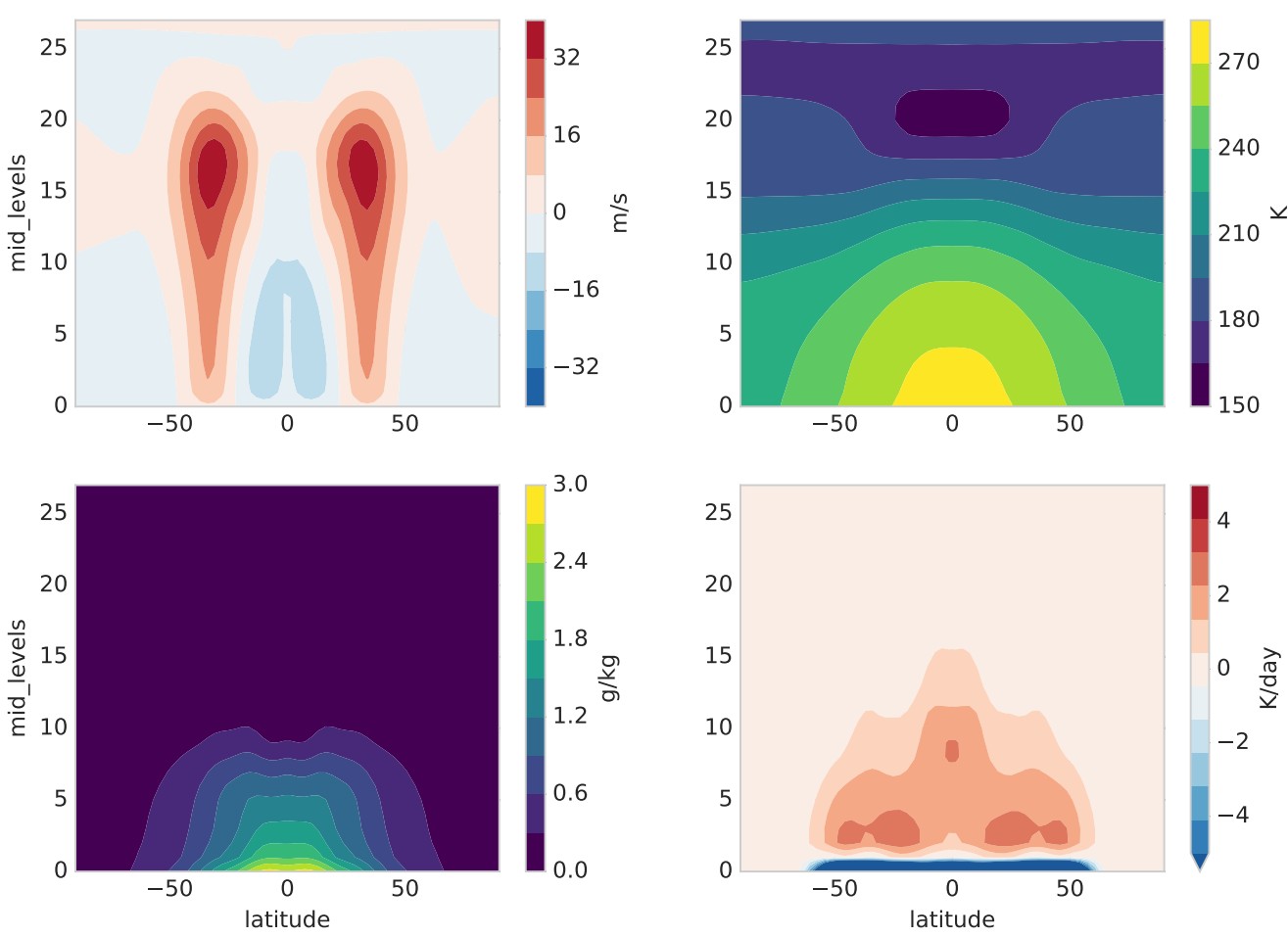

**Figure 9.** The zonal mean equilibrium profiles in the idealised GCM runs with no seasonal cycle. The plotted fields are, in clockwise order from the top left, the zonal winds, air temperature, convective heating rate and specific humidity respectively. The y-axis is in model levels and x axis is in degrees.

# 8 Conclusions and Future Avenues

`sympl` and `climt` represent a novel approach to climate modelling which provides the user with fine-grained control over the configuration of the model. `sympl` provides a rich set of entities which describe all functionality typically expected of a climate model. This set of entities (or classes) allows `climt` to be an easy to use climate modelling toolkit by allowing decisions about model creation and configuration to be made at a single location (the run script) and without ambiguity. The modular nature of the packages allows for code reuse as one traverses the hierarchy of models from single column model to three dimensional GCMs. We attempt to address concerns about plug-and-play type architectures (Randall, 1996) by ensuring the inputs and outputs of each model are cleanly documented, which makes it clear whether components are compatible or not. The use of Python allows for delegating computationally intensive code to compiled languages while still providing an intuitive and clean interface to the user. This choice also allows users access to a large variety of libraries written in Python for purposes ranging from machine learning to visualisation (Alpire, 2017).

The main focus in the near future would be to add more components, especially a cloud microphysics scheme, to allow `sympl`/`climt` to simulate a more realistic benchmark climate. Due to its flexibility, we believe our modelling framework is well suited to the simulation of general planetary atmospheres and for exoplanet modelling, and adding components relevant to these fields will also be a priority. Another important component to add would be a flexible grid interpolation component to allow interaction between components based on different model grids. While care has been taken to ensure that parallel computing is possible, we have yet to address the question of distributed memory and computing. While building models in a simple MPI scenario seems feasible in the near future, more sophisticated configurations with components running in parallel will need some thought and design.

Nevertheless, `sympl` and `climt` represent an important step towards creating flexible, usable and readable models. We hope that they will be a useful addition to the growing collection of Python based tools available to the climate science community.

*Code availability.* `sympl` is available at https://github.com/mcgibbon/sympl. The digital object identifier (DOI) for the version documented in this paper is https://zenodo.org/record/1346405.

`climt` is available at https://github.com/CliMT/climt. The digital object identifier (DOI) for the version documented in this paper is https://zenodo.org/record/1400103.

*Acknowledgements.* The first and third authors acknowledge a research grant from the Swedish e-Science Research Center (SeRC). The second author was funded by DOE grant DE-SC0016433 as a contribution to the CMDV (CM)4 project, and by the Natural Sciences and Engineering Research Council of Canada (NSERC) Postgraduate Scholarship Doctoral Program (PGS-D).

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
