# Peer review of "sympl (v. 0.4.0) and climt (v. 0.15.3) – Towards a flexible framework for building model hierarchies in Python"

_Geoscientific Model Development, 2018_

## Short Comment (SC1) · 4 May 2018

Authors should provide DOI's to the particular release, eg. using Zenodo from github releases, see https://guides.github.com/activities/citable-code/ for details. Data and scripts for the benchmark simulations should made available as supplements,

Lutz Gross GMD Executive Editor.

---

## Referee Comment (RC1) · S. Hill (Referee) · 25 May 2018

**Review of "sympl and climt — Towards a flexible framework for building model hierarchies in Python"**

Manuscript authors: Joy Merwin Monteiro, Jeremy McGibbon, Rodrigo Caballero

**Summary of manuscript**

The authors introduce the `sympl` and `climt` open-source python packages for creating models of Earth's climate (or those of other planets) and/or its subcomponents. `sympl` is a general purpose tool for formally linking and running these models, while `climt` is a specific instance of a model built using `sympl`.

**Summary of my review**

(Note that I deliberately did not review the source code of either package nor have I attempted to use either package, and I only briefly skimmed the documentation. I thought this would be useful for the sake of providing a review, since I suspect most readers will be in the same position.)

These kind of open-source, carefully designed, richly documented simulation and analysis tools are sorely needed in climate science and related fields, and I sorely hope that one day using them becomes standard operating procedure. I found especially compelling the prospect of changing a single column model to a full dynamical model with only a few lines of additional code.

However, my initial excitement quickly faded while reading as I found it very difficult to follow the manuscript. Still at this point, I am confused about some fundamental aspects of the packages. I think this could be addressed through the use of more concrete examples and some reorganization.

Therefore, though I look forward to this manuscript being published eventually, I recommend major revisions at present. My remaining comments are below.

Signed,
Spencer Hill
UCLA and Caltech

**Major Comments**

**Lack of concrete examples**

Most of the manuscript describes the packages in abstract terms, in such a way that I really struggled to understand what was meant much of the time. The concrete examples that are given are mostly not until the latter part of the manuscript, by which point I was already confused. I think it would be useful to move these earlier. This may require some considerable reorganization of the manuscript, but I think that would helpful. In general it's much easier for the reader to get a concrete example first, and then use that to discuss a more general principle.

Alternatively (or in conjunction with the existing examples), another potentially useful approach would be to create a toy example or two (or, if more, as few as possible) that are continually referred back to as concrete examples when more abstract issues are presented. At least one should use `sympl` without `climt`, while at least one should use `climt`.

For example, in Section 5, as described the `Prognostic` and `Implicit` types are so similar that it's not clear why they are treated separately. A specific example of each type would make that much clearer. And if the same examples were used repeatedly or gradually built out throughout the manuscript, it would facilitate the reader making connections across sections.

**Scope of `climt` not clear**

It would be easy to understand if `climt` was simply a particular model that was generated using the `sympl` framework. But instead, as I understand it, `climt` is itself a framework, albeit a more concrete one in which physical processes are explicitly represented, from which models can be generated. That may well be a useful approach (I suspect it is), but for the purposes of this manuscript, it makes understanding the packages more confusing.

For this reason, I reiterate the comment above that introducing a minimal, concrete example of a model developed using `sympl`, and separately one developed using `climt`, would be valuable.

Also, given that `climt` has been around for a long time, but previously in a very different guise (i.e. Rodrigo's `CliMT-legacy` Github repo), it would be worth noting this history somewhere.

**Platform independence**

The manuscript referred primarily to computing environments akin to institutional clusters or supercomputers, i.e. those with multiple processors (e.g. Section 3.3). But, especially for the simple models being espoused in the introduction, laptops and tablets can be sufficient. Separately, there is the coming era of web-based computation (e.g. Amazon Web Services; the Pangeo project). How do `sympl` and `climt` handle these different environments, what configuration etc. is required to go from one to another, etc.? Some discussion of this would be useful.

**Line-by-line and minor comments**

**Page 1**

**L1** Consider replacing "represent" with a stronger word like "are"

**L4 and elsewhere** Oxford comma; I like it (i.e. I would put a comma after "inter-operable", but it is ultimately personal preference.

**L4 and elsewhere** hyphen in "fine-grained"

**L7-11** I'm not sure listing the different data structures is useful in the abstract. Consider a more compact, higher-level description instead, more like the subsequent paragraph.

**L8** Implicit should be `code-formatted` like the other components.

**L13** I don't think that Cython is well-known enough to be referenced in the abstract without introduction. Consider just omitting this clause.

**L13** The meaning of "aims" is ambiguous. Does that refer to planned future work, or is it the package's philosophy/design principle?

**L14** I found "trade-off" confusing here. Is this a comparison amongst the "different APIs"? Or between the APIs collectively and something else? I think it's the former, but it still could be made clearer.

**L14 and elsewhere** consider replacing "which" with "that"

**L16** "is" -> "are" and needs another verb, e.g. "are performed"

**L17** hyphen in "Python-based"

**L17-19** I don't really understand this sentence. What is the connection between modularity and "using online data analysis"? And what does "using online data analysis" even mean? It just feels like a weak sentence to end on.

**L21** "mutually" redundant; consider omitting

**L22** comma after "chemosphere)"

**Page 2**

**L2** Omit "slightly"; it is sometimes the (fortunate) case that a very simple model very clearly elucidates the behavior of a much more complex one.

**L5-6** Part of the emphasis of the Jeevanjee et al paper was on the fact that there is no single hierarchy that is universally applicable. As such, I recommend against referring to "the" modeling hierarchy; consider "the specific model hierarchy suiting their needs." This is also in line with the packages' goal of enabling users to plug-and-play components as they see fit.

**L7** Consider "...this hierarchy remains a challenge..."

**L12-14** I am unfamiliar with ESMF, and it isn't clear from the text how it provides the fine grained control referenced.

**L15** hypen in "framework-based"

**L15** I would start a new paragraph here.

**L15-17** `sympl` and `climt` are being referred to before they have been introduced in the main body.

**L17-20** "we believe ... our design to be unique" is too vague to be convincing. What are the salient differences?

**L22-27** I would put this before the preceding sentences that refer to sympl and climt. As written, it jumps back and forth from motivation (modularity) to the packages, back to another motivation (reproducibility), back to the packages again.

**L30-32** If you choose to preview the manuscript in this way (which I support), it is better to be explicit about what is in each section. E.g. "...frameworks have to solve (Section 2), ..."

**Page 3**

**L15** the operate -> that operate

**L24** Section title uses "taxonomy", but text here (and elsewhere) uses "ontology". I would pick one or the other to keep down the jargon.

**L27** I don't understand this sentence, mainly because I don't know what is meant by "display". What does it mean to display different behaviors consistently?

**L29-30 and subsequent** example of where concrete examples would be extremely helpful

**Page 4**

**L5-6** I don't understand what point is trying to be made here. Examples would help.

**L10-19** Examples would help.

**L21-24** Here's a paragraph where, thankfully, an example *is* utilized. But I don't understand the example given. What does it mean to "build a full model" in this context? Outputting radiative tendencies *requires* some treatment of radiative transfer, however simple or complicated. Does "interact" simply mean "output results"?

**Page 5**

**L12-14** This feels out of place. The rest of the section is about general principles, not `sympl` and `climt`. Move to next section?

**L16-24** I don't see the relevance of these autobiographical/historical details. Consider opening this section more directly, with a brief summary of the design decisions that you will subsequently detail in the rest of the section.

**L23** "object orientation" -> "object oriented"

**L22-30** Examples would help.

**Figure 1**

- Names don't match those in Section 3

- Why does "Sympl+CliMT" bracket on the left not extend to the runscript?

- It is possible to reduce the number of intersecting lines in the bottom half; as presented here it feels somewhat unfair (the more intersections, the more complicated it looks).

**Page 7**

**L3** For xarray cite *Hoyer and Hamman* (2017)

**L6-7** Consider instead "This exposes the powerful analysis capabilities of `xarray`, allowing users..."

**L8-11** What is meant by low-level operations being "simple"? Simple for the user? What is the connection to memory layout? It also might help to put a paragraph break before this; it feels like a separate topic from the higher-level `xarray`-related stuff discussed above it.

**L12-17** This recommendation for how users should use the API feels out of place in the "Design Decision" section... it would seem to be better placed in the next section

**L26** Is `Diagnostic` used by `Prognostic` and `Implicit`? Since they both also refer to outputting diagnostics.

**L27-29** This has the start of a useful example re: the CFL criterion, but it's not detailed enough to be useful. In other words, even with this example, I still can't tell why `ImplicitPrognostic` is needed in addition to the other two.

**Page 8**

**L6-7** I don't understand this description of a `Wrapper`. Why are these needed in addition to the other components?

**L24** Probably worth mentioning that this is important for simulations starting at year 0 or 1 (or really long simulations) for which the limited range of numpy datetime supported dates otherwise becomes a real pain.

**Page 9**

**L3** Models don't have to have all three spatial dimensions, right?

**L9** I would omit "to perform sensitivity experiments". There are cases (including some of my own) wherein varying planetary parameters like rotation rate is the *main* focus, not a sensitivity test.

**Page 10**

**Fig. 3** A more descriptive caption would help. So would explicitly populating the `__init__` and `__call__` methods.

**Page 12**

**Fig. 4** Panel labels and a legend would help a lot. Also, I know I keep saying this, but I just don't really understand a lot of this figure (apart from panel b). Take panel (a): what is the meaning of the arrows from Current State and Diagnostics converging at a cross and then joining into one? Why are the two Prognostic boxes in panel (c) in their own black outlined box?

**L2-5** I think it would help to first state the main focus of `climt`, namely providing actual usable scientific components that can be used to generate physically meaningful models. Then mention the other peripheral aspects: configuration options, helper functions, additional attributes.

**L7** "typically" is confusing; doesn't a spectral core step a model forward in spectral space by definition?

**Page 13**

**Footnote 6** Consider omitting; doesn't seem important.

**L12** Why separate dimensions into the "core" ones and this `extra_dimensions` category?

**L20** What does "by hand" mean?

**L24** Is Section 7 supposed to be Section 6.3?

**L28-29** incomplete sentence

**Page 14**

**all** citations and/or URLs for these different projects/components would be nice (e.g. CDMIP, OpenMP). For the insolation, grid scale condensation, and ice modules, did you code these up from scratch? If yes this should be noted. If not their original source/predecessor should be cited.

**Page 15**

**L2 and elsewhere in paper** capitalization typo

**L10** regression tests are mentioned; are there unit tests also?

**Figure 5**

**L1** PEP8: lines 1-2 would fit on a single line

**L37** 60000 is a magic number; define as a constant i.e. `NUM_TIMESTEPS = 60000`

**L41** Don't understand what "This order is arbitrary" means in this context. Could have run Implicit first and then TimeStepper? As written, it's counterintuitive to time step first and then call the physics.

**L44** another magic number

**L49** Feels weird to increment the time by hand like this, when everything else is so boxed into the `sympl` data structures. Why doesn't `TimeStepper` do this?

**Caption** What code was omitted? If at all possible include all the code, simplifying the example if necessary; otherwise it feels like a black box.

**Page 17**

- It would be really nice if the output in Figure 6 came directly from the model shown in Figure 5. Is that the case?

L8 a idealized -> an idealized

L8-10 Can you provide the runscripts for each? Potentially as supplmental material?

**Page 18**

**L5-6** Don't understand this, and an instance c.f. major comment above that gives the impression that these tools aren't easily ported to non-HPC contexts

**Page 19**

**L8** There was little mention in the paper of the documentation.

**L19-21** Ending with this limitation doesn't do you any favors. Consider adding an additional sentence or two emphasizing that, despite this limitation, these tools are functional/valuabel/etc.

**`climt` **online documentation**

- A more welcoming main page would be helpful; xarray's docs (`http://xarray.pydata.org/en/stable/`) are a great model to follow.

**References**

Hoyer, S., and J. Hamman (2017), Xarray: N-D labeled Arrays and Datasets in Python, *Journal of Open Research Software*, *5*(1), doi:10.5334/jors.148.

---

## Referee Comment (RC2) · Anonymous Referee #2 · 26 May 2018

**General Comments:**

A journal article needs to be published on sympl and climt, to document the usefulness and importance of the models. Even though climt is one of the earliest models of its type, the design decisions that have characterized the model have proven to be robust. Sympl, while newer, shows similar promise in terms of usefulness and design.

The flow of the current paper, however, I found difficult to follow. It seems to me that the paper is trying to do two things at once, and the ways these two theses are pursued leave the paper not as strong and focused as it could be. On the one hand, the paper seems to provide a summary documentation or manual of the models. On the other

hand, the paper aims to provide an exploration of the implications of the structure and design of the models, as shown in the subtitle, "Towards a flexible framework for building model hierarchies in Python."

The difficulty I think is that these two purposes of the paper, as it is currently written, do not complement each other as well as they could. Put another way, the manual-like descriptions of the model can be more integrated into the exploration-of-implications aspect of the paper. That is, the descriptions of the model can better flow into the implications the authors are trying to bring out. Below are a few examples of this relative lack of connection:

- Section 4 provides a manual-like discussion of design decisions. This listing provides history and details the usefulness of Python as a language. But how does this connect to the thesis that the models provide a nice framework for traversing model hierarchies? I'm not saying that the features of Python listed do not contribute to such a traversal but rather that having described these desirable features, the paper appears to do little with describing how these features support their thesis. For instance, this could be done by providing some code snippets that illustrate how the broad ecosystem Python provides makes ML analysis of model results a piece-of-cake whether one is running a single column model or moist GCM and enables easy intercomparison between results at different levels of the hierarchy.

- Section 5 describes the abstractions sympl uses, but there does not appear to be much discussion of how these abstractions work to simplify traversing the model hierarchy, gain greater understanding of the model, or enable more or better science to be done. Section 5.5 provides a workflow and Figure 4 shows data flow, but it's not clear what the message is. Neither the body of the paper nor the caption tell us the message. Is the message that the data flow is easy to map unlike the traditional climate model spaghetti shown in the bottom of Figure

1? That the sympl abstractions in Section 5 enable more science to be done by making it easier to set variables? Explicitly making the connection between the model description and the implications, plus providing an illustration, would be helpful.

- It's unclear to me what is the purpose of Section 6.2 on the climt model composition. Is it to say climt will have additional features? How is this connected to the thesis of the paper vis a vis traversing hierarchies? The section says automation will result and the sympl components are used in this task but there is no other description of what automation means in this case or how that fits into the workflow of understanding and using a model.

- On p. 15, line 1, the authors mention that the difference between a single column model and moist GCM is only 40 lines. Given how little extra code is required, why do the authors not show the difference (maybe as a side-by-side comparison) and walk the reader through the structure and differences? This would be a very effective way of making the point of how easy it is to traverse the modeling hierarchy.

- On p. 19, the authors describe the novelty of their approach and the ability of using the models for "fine-grained control." However, there are no examples of experiments illustrating such use of the model; for instance, as suggested earlier, a side-by-side comparison of the code used and model results from a single column and moist GCM, done with a single script. Put another way, examples of this sort and the analysis it supports, as opposed to only benchmarking, differentiate model documentation from an exploration of what a model can do.

- In Section 10, the authors describe how easy it is to create new models for exoplanet research. Here too, an example showing the ease of either transitioning from an Earth model to an exoplanet model, or creating an exoplanet model from scratch, would help make the point.

Again, I want to emphasize that I think the message the authors have needs to be published, but I think the paper in the current form needs improvement. Perhaps the thesis the authors are advancing is obvious to users of sympl and climt, once a summary description of the model is given, but for the reader who lacks such experience, more explicit description of how the model's structure leads to the benefits claimed is needed for the paper to be as convincing as it can be.

**Specific Comments:**

- I was a little unclear on the purpose of Figure 1. What do the arrows mean? What's flowing between the boxes? Data? Control? As far as I can tell, neither the caption nor the text of the paper provides such a description.

- p. 12: More description could be made of how sympl and climt are related to each other. The only description I found was lines 2–5 on p. 12. I'd recommend putting a description like that earlier in the paper and providing a more detailed description of what that means later on.

- Another source the authors might want to consider referencing regarding software frameworks for modeling is Schmitz (2014)'s, even though that work focuses on hydrological models: https://dspace.library.uu.nl/handle/1874/293818. Regarding issues of component ordering and a framework to manage subroutine execution order (p. 4, lines 16–24), the authors might want to consider Lin (2009)'s work: https://www.geosci-model-dev.net/2/1/2009/gmd-2-1-2009.html.

**Technical Corrections:**

- p. 3, line 15: Should be "The processes that operate within each . . . ."

- p. 13: I understood older versions of climt had Federations, but the text says they are to be created. Am I mistaken? I'm not saying the authors necessarily need to correct this, but it was something I noted.
- p. 17: Are the Figure 7 results generated by the Figure 5 script? If not, can the authors describe where to find the script to generate those results?
* * *

---

## Author Comment (AC1) · 20 Jul 2018

**Response to the reviews and short comments for the manuscript "sympl (v. 0.3.2) and climt (v. 0.11.0) – Towards a flexible framework for building model hierarchies in Python"**

We thank both reviewers for their fairly extensive and insightful comments. The main issues raised by both reviewers has lead to an extensive re-organisation of the manuscript. We hope that the new version will be better organised, more readable and engaging.

**1 Common issues raised by reviewers**

**1.1 Lack of examples**

We have addressed this issue by replacing the current section 2, which presents an abstract discussion about modelling frameworks with a new section which presents a set of examples which try to accomplish the following –

- Show the user how to build physically useful models

- Show how easy it is to change a single column model to a GCM

- Show the steps required to write a `sympl` component

Spencer mentioned in his review that it might be useful to build a model using only `sympl` without `climt`. However, given that `sympl` does not aim to provide scientific components, we felt that such an example would be too simplistic to be of real interest to readers. We prefer instead to present the layout of a `sympl` component and emphasise how easy it is to write such a component.

**1.2 Mixing conceptual issues with API description**

The new organisation first presents a quick user guide to build models. We then discuss the conceptual issues (what is currently most of Section 2) in a separate section. This allows to the reader to first see how easily scientifically useful models can be built, and then read about the reasons why such ease is possible with `sympl` and `climt`.

To address the issue that the conceptual and manual-like parts of the paper do not complement each other (raised by the anonymous reviewer), we have rewritten the conceptual parts of the paper to refer frequently to the examples presented in the new Section 2. We hope this will help make the design decisions clearer since every such description of a design decision will now refer to actual line/lines of code.

**1.3   Current scope of `climt`**

As it currently stands, `climt` is simply a collection of `sympl` compatible components (a toolkit). The framework-like features of earlier versions of climt, encapsulated in the `Federation` object, have not yet been implemented in the current version. This has also been noted in the previous Figure 1 of the manuscript. We are still mulling over the design of `Federation` given the new structure of `climt`, and thank the anonymous reviewer for pointing us to relevant references.

**1.4   Figure 1 and 4 unclear**

**1.4.1   Figure 1**

- We have fixed the balloon corresponding to Computing resource configuration to reflect the terminology in Section 3.

- We have renamed "Behavioural Configuration" to "Interfacial Configuration" to emphasise the fact that we only modify quantities at the interface (input/output) of a component. This change has been made in the text as well.

- We have lengthened the curly bracket to an appropriate size.

- From the very outset, we had tried to reduce the number of intersections of the lines for exactly the same reasons that Spencer mentioned. This was the best we could do.

- The caption has been modified to highlight the fact that the arrows correspond to control or responsibility, i.e, what file or module is typically responsible for a particular kind of configuration.

**1.4.2   Figure 4**

- We have redrawn the figure to make it clearer

- We have expanded the captions substantially explaining what each panel means.

**1.5 Code availability**

The model scripts to produce the results in the paper will be added as supplementary material during this revision. We have also generated DOIs for both packages, which will be incorporated into the manuscript. These changes also address similar issues raised by Lutz Gross in his short comment.

**2 Issues raised by Anonymous Reviewer**

**2.1 Link between features/design decisions and objective (traversing model hierarchies) unclear**

We thank the reviewer for pointing this out. We intend on addressing this issue by presenting two examples, a radiative-convective equilibrium model and a moist GCM in Section 2 and referring back to these examples frequently in subsequent sections where we describe features and design decisions of both packages.

We hope this strategy will achieve two things – illustrate how easy it is to move from a single column model to a 3D model, and what design decisions make it easy to do so.

**3 Issues raised by Spencer Hill**

We thank Spencer for his detailed and constructive comments. His suggestions for restructuring the paper were definitely very useful in making this manuscript as accessible as `sympl` and `climt` hope to be!

**3.1 Platform Independence**

Though we have not tested our packages on AWS explicitly, we do not forsee any obvious difficulties, since wheels are available for both packages. In fact, we have successfully been able to use `climt` on the web-based Google Colaboratory notebooks. We will make this clear in the revised manuscript.

**3.2 Other comments**

Most of the line-by-line comments have been addressed either by making appropriate changes or deleting sentences which could potentially confuse the reader.

**4 Changes made to `sympl` and `climt` since the time of submission**

Since the time manuscript was submitted, we have made some changes to both packages, which are reflected in the manuscript. The major changes are:

- Renaming the basic classes in `sympl`. Feedback from some of our users led to renaming the basic classes to clarify what they actually are. Also, we realised that `TimeStepper` was simply a kind of `Implicit`. Thus, the following changes were made:

  - `Prognostic` → `TendencyComponent`
  - `Diagnostic` → `DiagnosticComponent`
  - `Implicit` → `Stepper`
  - `TimeStepper` → `TendencyStepper`
  - `ImplicitPrognostic` → `ImplicitTendencyComponent`

- `sympl` classes have been restructured to make them more than just a basic description. In particular, the state dictionary is no longer passed onto the component code directly. Instead, a set of consistency checks are performed, the numpy arrays are extracted with the correct units, and these raw numpy arrays are passed onto the scientific code. This eliminates the requirement for developers to write boilerplate code for consistency checks and array extraction. This change has also been made in the corresponding section in the manuscript.

- `climt` has also been refactored to reflect these changes in `sympl`, resulting in a much cleaner API. However, there is very little if any change to how the user interacts with `climt`.

- These changes have resulted in new releases of both packages. Hence, the version numbers in the title will also be changed.

---

## Author Response (AR1)

**Letter to the Editor:**
**Changes in the manuscript**

The manuscript has been overhauled completely to meet the comments of the reviewers. In particular, the second section has been changed completely, and now presents a series of examples which show how to traverse a model hierarchy using Sympl and Climt. We then move on to design considerations and decisions (now a single section created by merging the previous design considerations and design decisions sections), where we frequently refer to the examples in the previous section to illustrate the issues we are trying to solve using these packages. These examples eliminated the need to present an example at the end of the manuscript.

The other major change was that Sympl and Climt were updated during the review process, and this is reflected in the names of the Sympl classes and version numbers in the paper title. Some minor API changes were also documented.

Furthermore, the line-by-line comments of the reviewers for the previous version of the manuscript have been addressed before any changes were made.

Other changes that were made to the manuscript are:

- Zenodo DOIs were added for both packages.

- The figure captions were overhauled for most figures.

- The ordering in Figure 6 was changed.

- Two panels were removed from Figure 8 since they did not add much information.

- The scripts used to run the benchmark simulations have been added as supplementary material

Sincerely,
Joy, Jeremy and Rodrigo

[revised manuscript text omitted]